# Learning and Investigation of the Role of Angiotensin-Converting Enzyme in Radiotherapy for Nasopharyngeal Carcinoma

**DOI:** 10.3390/biomedicines11061581

**Published:** 2023-05-30

**Authors:** Yanan Ding, Huanhuan Xiu, Yanling Zhang, Miaola Ke, Letao Lin, Huzheng Yan, Pan Hu, Meigui Xiao, Xu He, Tao Zhang

**Affiliations:** 1Department of Otolaryngology, The First Affiliated Hospital of Jinan University, Jinan University, Guangzhou 510630, China; 1234a@stu2019.jnu.edu.cn; 2Department of Anesthesiology, The First Affiliated Hospital, Sun Yat-sen University, Guangzhou 510062, China; xiuhh3@mail.sysy.edu.cn; 3School of Laboratory Medicine and Biotechnology, Southern Medical University, Guangzhou 510515, China; 4Department of Blood Transfusion, State Key Laboratory of Oncology in South China, Sun Yat-sen University Cancer Center, Guangzhou 510060, China; keml@sysucc.org.cn; 5Minimally Invasive Interventional Division, State Key Laboratory of Oncology in South China, Sun Yat-sen University Cancer Center, Guangzhou 510060, China; linlt@sysucc.org.cn (L.L.); hupan@sysucc.org.cn (P.H.); xiaomg@sysucc.org.cn (M.X.); 6Department of Interventional Radiology, The Third Affiliated Hospital, Sun Yat-sen University, Guangzhou 510630, China; yanhzhdr@163.com; 7Interventional Medical Center, Zhuhai People’s Hospital, Zhuhai 519050, China; hexu@163.com

**Keywords:** health informatics, nasopharyngeal carcinoma, ionizing radiation, angiotensin-converting enzyme, reactive oxygen species

## Abstract

Ionizing radiation (IR) is an important treatment for nasopharyngeal carcinoma (NPC) that mainly kills tumor cells by producing large amounts of reactive oxygen species (ROS). Intracellular ROS levels affect the sensitivity of tumor cells to IR. Recently, angiotensin-converting enzyme inhibitors (ACEIs) and angiotensin-converting enzyme (ACE) have been found to affect the intracellular levels of ROS. Therefore, we performed a health informatics assessment of ACE in the TCGA database. We explored the effect of ACE in NPC cells. We found that either knockdown of ACE or inhibition of ACE by enalaprilat could decrease ROS levels in NPC cells. Furthermore, knockdown of ACE or inhibition of ACE by enalaprilat could reduce IR-induced ROS levels. ACE knockdown or inhibition reduced IR-induced DNA damage and apoptosis. ACE overexpression increased the level of ROS in NPC cells and further increased sensitivity to IR. These findings indicate that ACE influences the effect of IR by regulating the level of ROS in NPC cells.

## 1. Introduction

Ionizing radiation has long been a pillar of tumor treatment, especially for solid tumors [1]. The biological effects of ionizing radiation are mainly caused by the formation of ROS [2], which causes damage to DNA [3,4] and leads to cell death. Although IR is commonly used to treat tumors, resistance to IR is one of the main reasons for the failure of radiation therapy [5,6]. In recent years, it has been found that radioresistant tumor cells have lower levels of ROS [7,8,9].

Angiotensin-converting enzyme is a metallopeptidase that belongs to the gluzincin family of metalloproteases [10]. As is well known, angiotensin-converting enzyme 2 (ACE2) is the receptor of COVID-19 (SARS-CoV-2). While the sequence of ACE is similar to ACE2, its function can be understood as opposite to and competing with ACE2. Recent studies have shown that the overexpression of ACE increases intracellular ROS levels [11]. Renin-angiotensin system blockers, a class of drugs known as angiotensin-converting enzyme inhibitors, are clinically recommended [12]. Evidence suggests that angiotensin-converting enzyme inhibitors can reduce the level of ROS and enhance endogenous antioxidant defense [13,14]. 

Nasopharyngeal carcinoma, a malignant tumor derived from the nasopharyngeal epithelium, has particularly high prevalence in Southeast Asia and Southern Asia [15]. Because of its high radiosensitivity, radiation therapy remains an effective form of treatment for nasopharyngeal carcinoma [15,16]. However, there remains a subset of patients whose tumors do not benefit from treatment with radiotherapy due to radioresistance [17,18]. Hence, it is crucial to explore the mechanisms of resistance to radiotherapy for NPC and to develop radiosensitization strategies for antitumor therapies. 

We first performed a health informatics assessment of ACE in the TCGA database. Whether ACE play roles in IR in NPC is unclear. Therefore, in our study, we used flow cytometry to determine the effects of ACE and ACE inhibition on ROS levels in NPC cells. In addition, we determined whether ACE plays a role in NPC radiation therapy by measuring cell proliferation, cell apoptosis, and DNA damage. To the best of our knowledge, this study is the first to assess the role of the ACE in the radiation treatment of NPC cells.

## 2. Materials and Methods

### 2.1. Bioinformatics Analysis

For Gene Expression Omnibus (GEO) datasets, mRNA expression data were downloaded and analyzed using HOME for Research (https://www.home-for-researchers.com). Analysis of the Cancer Genome Atlas (TCGA) dataset was performed using Xiantao Academic Online Tools (https://www.xiantao.love/products, accessed on 20 March 2023)). The correlation between overall survival (OS) and ACE expression was examined using Kaplan–Meier (K-M) survival plots according to the Kaplan–Meier Plotter database (http://kmplot.com, accessed on 4 April 2023).

### 2.2. Cell Culture and Transfection

The human NPC cell lines CNE1 and CNE2 were obtained from Sun Yat-sen University Cancer Center. Cells were grown in 1640 medium (Gibco, Grand Island, NY, USA) supplemented with 10% fetal bovine serum and 1% penicillin–streptomycin (Life Technologies, Grand Island, NY, USA) in a humidified incubator with 5% CO_2_ at 37 °C.

ACE was overexpressed and downregulated by transiently transfecting ACE plasmids or ACE siRNA, respectively. The ACE overexpression plasmids and the control pcDNA (empty vector) were obtained from HanBio (Shanghai, China). The small interfering RNA (siRNA) used to knockdown ACE and negative control were purchased from RiboBio (Guangzhou, China). The specific sequences targeting ACE are listed as follows: siACE-1#, GAGGCCAACTGGAACTACA; siACE-2#, CTAGCCCTCTCAGTGTCTA. Lipofectamine 3000 (ThermoFisher, Waltham, MA, USA) was used to transfect the overexpression plasmids according to the manufacturer’s protocol.

### 2.3. Cell Viability Assay 

The activity of the cells was detected using CCK8 reagent (Dojindo, Kumamoto, Japan). Nasopharyngeal carcinoma cells were cultured in 96-well plates with 1500 cells per well and 100 μL of culture medium. For overexpression or knockdown cells, cells transfected with ACE plasmids or siRNA and their corresponding controls were inoculated in 96-well plates (1500 cells per well) and cultured overnight. Subsequently, the cells were treated with a radiation dose of 4 Gy; control cells were not irradiated. 

Enalaprilat (Abmole, Houston, TX, USA) was dissolved in DMSO and diluted to different concentrations according to the manufacturer’s instructions [19,20]. For cells requiring ionizing radiation, NPC cells (1500 cells/well) were treated with enalaprilat for 12 h followed by radiotherapy, and DMSO was used as the control treatment.

Next, the medium was removed and the cells were washed twice with PBS. Medium (90 μL) and CCK8 (10 μL) were added to each well, then the cells were incubated for 2 h at 37 °C. A microplate reader spectrophotometer was used to measure the optical density (OD) 450 nm.

### 2.4. Apoptosis Analysis

Annexin V-FITC/PI (Keygen, Nanjing, China) was used following the manufacturer’s instructions. CNE1 and CNE2 cells were seeded in 6-well plates. For cells in which ACE was overexpressed or knocked down, radiotherapy was performed 48 h after the completion of transfection. For drug-treated cells, radiotherapy was performed 12 h after treatment with enalaprilat or DMSO. The dose of radiotherapy was 4 Gy, and cells were collected 48 h after radiotherapy. A CytoFLEX flow cytometer (Beckman, Brea, CA, USA) was used to detect apoptotic cells.

### 2.5. Immunofluorescence Staining

Cells were inoculated in co-focal dishes. After 2 h of cellular ionizing radiation, the cells were washed twice with PBS and then fixed in 4% paraformaldehyde (Sigma Aldrich, Burlington, MA, USA) for 15 min followed by permeabilization with 0.5% Triton X-100 (Thermo Fisher, Waltham, MA, USA) and blocking with 5% BSA (Sigma Aldrich, Shanghai, China) for 1 h. The cells were then incubated overnight at 4 °C with primary antibodies targeting γh2AX (CST 80312, 1:400 dilution) and/or 53BP1 (Beyotime AF6111, 1:400 dilution). After three washes with PBS, the cells were incubated for 1 h at room temperature using secondary fluorescent antibodies (Cy3 or FITC, Beyotime, Haimen, China). Nuclei were stained with 4’,6-diamidino-2-phenylindole (DAPI) (Keygen, Nanjing, China) for 3 min and immediately visualized using a confocal laser scanning microscope (Olympus FV1000, Tokyo, Japan).

### 2.6. Alkaline Comet Assay

A comet assay was performed to determine the extent of oxidative DNA damage, following the manufacturer’s instructions (Trevigen, Gaithersburg, MD, USA). Cells were exposed to radiation doses of 4 Gy. After 2 h, the cells were harvested. The cell suspension was mixed with 50 μL of low melting agarose (LMA) at 37 °C and spread on a slide. After gel formation, the slides were submerged in alkaline lysis buffer for 1 h at 4 °C and kept in darkness. Next, the slides were placed in freshly prepared electrophoresis buffer (200 mM NaOH and 1 mM EDTA, pH >13) at 4 °C for 30 min to allow for the unwinding of DNA and alkali-labile damage sites. Subsequently, the samples were electrophoresed at 21 V for 30 min at 4 °C. After electrophoresis, the slides were washed three times for 5 min at 4 °C in dH_2_O. Slides were viewed under a fluorescence microscope (Olympus, Tokyo, Japan). CASP software was used to calculate the length of the appendix. The tail moment was used as a measure of DNA damage using the following formula: tail moment = fraction of DNA in the tail × tail length.

### 2.7. Western Blotting

The cells were collected, lysed in lysis buffer (containing 0.1% protease inhibitor, 0.5% 100 mM PMSF, and 1% phosphatase inhibitor) and then centrifuged at 12,000 rpm for 15 min at 4 °C. The supernatant was collected and the total protein content was determined using the BCA method (Thermo Fisher, Waltham, MA, USA). Equal amounts of protein were separated by 7.5–12.5% SDS-PAGE and transferred onto 0.25 nm PVDF membranes. The membranes were then blocked with 5% skim milk powder and incubated overnight at 4 °C with primary antibody. After washing with Tween 20 (TBST) (Sangon Biotech, Shanghai, China), the blots were incubated with the corresponding secondary antibody (Cell Signaling Technology, Beverly, MA, USA) at 37 °C for 1 h. After washing again with Tween 20 (TBST), the target bands were visualized by enhanced chemiluminescence (4Abio, Beijing, China) using a chemiluminescence imager (Bio-Rad, Hercules, CA, USA). The primary antibodies used in this study were γH2AX (CST 80312, 1:1000 dilution), H2AX (CST 80312, 1:1000 dilution), GAPDH (CST 5174, 1:1000 dilution), and ACE (Affinity AF5197, 1:1000 dilution).

### 2.8. Flow Cytometry Analysis of Intracellular ROS

A DCFH-DA fluorescent probe (Beyotime, Shanghai, China) was used to assess intracellular ROS levels [21]. Briefly, cells were exposed to 4 Gy of radiation after 24 h. After exposure, the cells were incubated in the dark with DCFH-DA at 37 ℃ for 30 min. The cells were then washed three times with PBS, digested and analyzed for ROS levels using a CytoFLEX flow cytometer (Beckman, Brea, CA, USA).

### 2.9. Quantitative Real-Time Polymerase Chain Reaction (RT-qPCR)

Total RNA was extracted from NPC cell lines using an RNA-quick Purification Kit (Yishan Biotechnology, Shanghai China). Total RNA was converted to cDNA using Premix (Accurate Biotechnology, China). Quantitative real-time PCR (RT-qPCR) was performed using an Applied Biosystems Fast Real-Time PCR System (Roch, Ypsilanti, MI, USA). We used the 2-ΔΔCt method to determine how much the target gene expression changed compared to the GAPDH housekeeping gene control. The sequences of *RT-qPCR* primers were as follows: ACE Forward—TCTTCGCGCAGAGCTACAAC; ACE Reverse—TTTGGCGGTGGAGTAGATCCT; GAPDH Forward—AGAAGGCTGGGGCTCATTTG; GAPDH Reverse—GCAGGAGGCATTGCTGATGAT; β-actin Forward—CCATGTACCCAGGCATTGCT; β-actin Reverse—AGAACTTTGGGGGATGTTTGC.

### 2.10. Mitochondrial Membrane Potential Measurement

Mitochondrial membrane potential (MMP) in NPC cells was assessed using an MMP assay kit with JC-1 according to the manufacturer’s protocols (Beyotime, Shanghai, China). At low membrane potentials JC-1 forms JC-1 monomers, while at higher potentials JC-1 forms aggregates and displays red fluorescence. The transition from JC-1 aggregates (red fluorescence) to JC-1 monomers (green fluorescence) was observed using laser confocal microscopy to assess the reduction in MMP.

### 2.11. X-ray Irradiation

An X-ray irradiating instrument (Rad Source Technologies, Buford, GA, USA) was used to perform X-ray irradiation at Sun Yat-sen University Cancer Center. The dose rate was 1.78 Gy/min. Single irradiation doses were 4Gy.

### 2.12. Statistical Analyses

Statistical analyses were performed using SPSS software version 26.0, including independent-samples *t*-tests and one-way ANOVA. The specific statistical analyses are described in the legend. Data are shown as mean ± standard deviation (SD). Each experiment was repeated thrice. Statistical significance levels were set at * *p* < 0.05, ** *p*< 0.01, *** *p* < 0.001, and **** *p* <0.0001.

## 3. Results

### 3.1. ACE Expression Is Upregulated in NPC Patients

ACE2 is the receptor of COVID-19. While the sequence of ACE is similar to ACE2, its function can be understood as opposite to and competing with ACE2. It is well known that tissues with high levels of ACE2 expression are thought to be susceptible to direct SARS-CoV-2 infection. We performed a health informatics assessment of ACE2 using the head and neck squamous cell carcinoma dataset from the TCGA database. There were no significant differences in ACE2 expression between cancer tissues and normal tissues (Appendix A). Thus, we shifted our gaze towards ACE.

To determine the role of ACE in NPC, we first analyzed the available human NPC datasets from the GEO database (GEO: GSE53819) and found that the ACE mRNA level was upregulated in NPC tissues compared with normal nasopharyngeal epithelial tissues (Figure 1A). As a result of the small sample size in the GEO database and the lack of specific NPC samples in the TCGA database, we analyzed the mRNA expression levels of ACE in head and neck squamous cell carcinoma (HNSCC) from the TCGA database. There was a significantly higher expression of ACE mRNA in HNSCC tissues compared to normal tissues (Figure 1B). We further isolated 43 paired samples, tumor tissues, and non-tumor adjacent tissues from HNSCC, obtaining similar results (Figure 1C). To assess the prognostic value of ACE in HNSCC, we performed Kaplan–Meier survival analysis based on the TCGA database. Patients were split into two groups, an ACE high expression group and an ACE low expression group, finding that the overall survival times of HNSCC patients with high expression of ACE were longer than the patients with low ACE expression (Figure 1D). We further performed a Kaplan–Meier analysis according to the mRNA expression of ACE in HNSCC using the KM-Plotter database (Figure 1E–G). High expression of ACE predicted a longer survival time for Grade 1 tumor patients, while high expression of ACE did not increase overall survival rate for tumor Grades 2 and 3.

### 3.2. ACE Knockdown Reduces IR-Induced ROS Levels

To investigate the role of ACE in IR, we used the NPC cell lines CNE1 and CEN2 in follow-up experiments. ACE expression was reduced by small interfering RNA (siRNA) treatment of the NPC cell lines CNE1 and CNE2. RT-qPCR and western blot results demonstrated that siRNA effectively interfered with and reduced ACE expression (Figure 2A,B).

We measured the proliferation ability of CNE1 and CNE2 cells with or without ACE knockdown after IR. The results showed that ACE knockdown had no effect on the proliferation abilities of the CNE1 and CNE2 cells that were not exposed to IR (Figure 2C,E). However, after exposure to 4 Gy IR, the proliferation ability of these cell lines significantly increased after ACE knockdown (Figure 2D,F). To assess the effect of ACE knockdown on intracellular ROS levels, we measured the intracellular fluorescence intensity by flow cytometry using DCFH-DA fluorescence staining. The results showed that ACE knockdown decreased the intracellular levels of ROS without radiotherapy, and the ACE knockdown group had lower levels of ROS in the irradiated cells than the control group (Figure 2G–J).

### 3.3. ACE Knockdown Decreases Radiosensitivity in NPC Cells

ROS produced by tumor cells after stimulation by radiotherapy can directly damage mitochondria, resulting in a decrease in mitochondrial membrane potential [22]. Decreased mitochondrial membrane potential is both a marker of early apoptosis and a cause of DNA damage. To further determine the role of ACE in radiotherapy, we used JC-1 to evaluate changes in mitochondrial membrane potential after ACE knockdown. JC-1 labels mitochondria with high membrane potential using red fluorescence (JC-1 aggregates) and mitochondria with low membrane potential using green fluorescence (JC-1 monomers). JC-1 staining showed decreased MMP after irradiation, as indicated by higher JC-1 monomer intensity (green fluorescence), while ACE knockdown reversed membrane depolarization (Figure 2K). 

Cell apoptosis is one of the major mechanisms that induces cell death. We used Annexin V-FITC/PI double-labeled flow cytometry analysis to detect the rate of apoptosis in NPC cell lines with or without ACE knockdown after radiotherapy. We found that there were no differences in the apoptosis rates between the ACE knockdown and control groups in the absence of IR (Figure 2L–O). However, significant differences in the apoptosis rates were seen between the two groups with IR treatment at doses of 4 Gy. ACE knockdown decreased the rate of IR-induced cell apoptosis. These results suggest that ACE knockdown can decrease the sensitivity of NPC cells to IR.

### 3.4. ACE Knockdown Reduces IR-Induced DNA Damage

Ionizing radiation generates ROS in the exposed cells that attack DNA, including DNA single-strand breaks (SSBs) and double-strand breaks (DSBs) [4,23]. In DNA double- and single-strand breaks, phosphorylated histone H2AX (γH2AX) was detected and used as a surrogate marker for DNA lesions [24]. The colocalization of γH2AX and 53BP1 lesions represents the DSBs [25]. γH2AX foci and 53BP1 foci were detected by immunofluorescence (Figure 3A,B). As shown in Figure 3C,D, ACE knockdown significantly reduced the number of γH2AX and 53BP1 colocalization foci in the irradiated cells compared to controls.

In addition, anti-γH2AX antibodies were used in western blotting analyses to detect the levels of γH2AX expression (Figure 3E,F). As expected, IR induced the expression of γH2AX and ACE knockdown inhibited γH2AX expression in the irradiated cells.

DNA breaks can be detected using a comet assay [26]. In comet experiments, the percentage of DNA in the comet tail is proportional to the levels of DNA strand breaks [27]. We used an alkaline comet assay to test the DNA breaks. The comet tail moment (tail moment = fraction of DNA in the tail × tail length) is proportional to the level of DNA strand breaks [27]. As shown in Figure 3G–J, tail moment averages were not different between the ACE knockdown group and the control group at 2 h in the absence of IR. However, in the irradiated cells ACE knockdown significantly decreased the comet tail moment compared with the control group. These results suggest that ACE knockdown can reduce radiation-induced DNA damage.

### 3.5. Inhibition of ACE by Enalaprilat Reduces Levels of Endogenous ROS in NPC Cells

Having shown that ACE knockdown reduces the radiosensitivity of NPC cells, we continued to explore the effect of ACE inhibition by chemical inhibitors on NPC radiotherapy. Enalaprilat is an angiotensin-converting enzyme inhibitor that is commonly used in clinical practice, and which mainly inhibits the activity of ACE [28]. To investigate the role of ACE inhibition in IR, we used the ACE inhibitor enalaprilat in a follow-up experiment. We examined the effect of enalaprilat on the viability of NPC cells at concentrations ranging from 0 to 4 μg/mL via CCK8 assay. After 24, 48, 72, and 96 h, the OD450 nm value was measured to evaluate cell proliferation. As shown in Figure 4A, the CCK8 assay showed that enalaprilat with the concentration of 0–4 μg/mL had no significant impact on the proliferation of CNE1 and CNE2 cells.

ACE inhibitors are known to play a role in the regulation of intracellular ROS. We used flow cytometry to detect the effect of enalaprilat on intracellular ROS levels in NPC cells. According to the results, there were no changes in cellular ROS at enalaprilat concentrations of 0.5 μg/mL or 1 μg/mL, while a decrease in intracellular ROS occurred when the concentration increased to 1.5 μg/mL and continued to decrease when the concentration reached 2 μg/mL (Figure 4B,C). Because enalaprilat induced a reduction in reactive oxygen species levels in nasopharyngeal carcinoma cells at a concentration of 1.5 μg/mL, we selected this concentration for the follow-up study.

Having shown that enalaprilat reduces the level of ROS in NPC cells, we continued to explore whether there was any effect on ROS levels after radiotherapy. We found that enalaprilat decreased ROS levels in NPC cells (Figure 4D–G). After IR, enalaprilat suppressed the level of ROS in NPC cells. This phenomenon is consistent with ACE knockdown. 

### 3.6. Inhibition of ACE by Enalaprilat Reduces Apoptosis in NPC Cells after IR

Previous studies have demonstrated that low ROS levels contribute to the radioresistance of cancer cells [7,8]. Consequently, we examined cell viability using the CCK8 assay to assess the effect of enalaprilat on NPC cells with IR. We found that enalaprilat did not affect the viability of CNE1 and CNE2 cells without IR, while it promoted the proliferation of NPC cells compared with the control group after treatment with IR at a dose of 4 Gy (Figure 4H,I). Apoptosis caused by IR is one of the most significant effects of tumor radiotherapy. To further investigate the role of enalaprilat in NPC cells, we used flow cytometry to detect cell apoptosis with or without IR (Figure 4J,L). As shown in Figure 4K,M, the rate of apoptosis did not significantly differ between the enalaprilat and control groups in the absence of IR; however, after treatment with IR, the rate of apoptosis was lower in the enalaprilat group than in the control group. These results reveal that the ACE inhibitor enalaprilat in combination with radiotherapy reduces the sensitivity of NPC cells to IR.

### 3.7. Inhibition of ACE by Enalaprilat Reduces IR-Induced DNA Damage

We performed immunofluorescence with γH2AX antibodies to detect DNA damage foci (Figure 5A,B). The difference in the number of γH2AX foci in unirradiated cells between the enalaprilat group and the control group was not significant (Figure 5C,D). However, the ACE inhibitor enalaprilat group showed lower induced γH2AX levels than the control group at 2 h with IR. At the same time, western blotting was used to determine the expression level of γH2AX (Figure 5E,F). The results showed that the γH2AX protein levels were not different between the enalaprilat and control groups in the absence of IR. However, the γH2AX protein level was significantly decreased in the enalaprilat group at 2 h after IR exposure.

Next, a comet assay was used to detect DNA damage (Figure 5G–J). As shown in Figure 5G,I, In the absence of IR the comet tail moment was no different between the enalaprilat and control groups. After treatment with IR, the comet tail moment was shorter in the enalaprilat group than in the control group. These results suggest that the ACE inhibitor enalaprilat with radiotherapy leads to reduced DNA damage by reducing intracellular levels of ROS.

### 3.8. Increased Levels of ROS in NPC Cells Due to ACE Overexpression Lead to Enhanced Sensitivity to Radiotherapy

We previously observed that knockdown and inhibition of ACE can reduce ROS levels. Next, we overexpressed ACE in NPC cells. ACE was upregulated in the NPC cell lines CNE1 and CNE2 via transfection with pcDNA-ACE for overexpression. The overexpression of ACE mRNA and protein was confirmed by RT-qRCR and Western blotting, respectively (Figure 6A,B). We explored the effect of ACE overexpression on ROS levels. Flow cytometric analysis revealed that ACE overexpression increased intracellular ROS levels, and the ACE overexpression group had higher ROS levels than the no overexpression group in the irradiated cells (Figure 6C–F).

A CCK8 assay was used to detect the viability of NPC cells with or without IR (Figure 6G). The data showed that ACE overexpression had no effect on the proliferation abilities of CNE1 and CNE2 cells not exposed to IR. However, after IR treatment, ACE overexpression significantly reduced the ability of cells to proliferate. Next, we used Annexin V-FITC/PI double-labeled flow cytometry analysis to detect the rate of apoptosis (Figure 6H,J). The results showed that the percentage of apoptotic cells did not significantly differ between the ACE overexpression group and the control group in the nonirradiated cells (Figure 6I,K). However, the percentage of apoptotic cells in the ACE overexpression group was significantly higher than that in the control group in the irradiated cells. These results demonstrate that ACE overexpression increases the sensitivity of NPC cells to IR.

### 3.9. ACE Overexpression Enhances IR-Induced DNA Damage

The colocalization of γH2AX and 53BP1 lesions represents the DSB [25]. γH2AX foci and 53BP1 foci were detected by immunofluorescence. ACE overexpression significantly increased the number of γH2AX and 53BP1 colocalization foci at 2 h post-irradiation compared to the control group (Figure 7A–D). As shown in Figure 7E–H, there were no significant differences in the average comet tail moment between the ACE overexpression group and the control group in the absence of IR. In contrast, the comet tail averages were significantly higher in the ACE overexpression group than in the control group at 2 h after 4 Gy with IR. These results demonstrate that ACE overexpression increases radiation-induced DNA damage. 

## 4. Discussion

Radiation therapy is a common tumor treatment in clinical practice, inducing tumor cell death through DNA damage [29,30,31]. DNA damage caused by IR may occurs through direct damage, but mainly occurs indirectly through the formation of reactive oxygen species [32]. For indirect effects, radiation ionizes water to form free radicals and free electrons, leading to a cascade of chemical reactions and the generation of large amounts of ROS, which attack DNA and other cellular molecules, including proteins and lipids [3,4]. Approximately 80% of cellular components consist of water, which plays a leading role in IR-induced biological effects [33]. Compared to normal cells, tumor cells have higher levels of ROS [34,35,36], and are highly sensitive to agents that induce further ROS stress [37]. Hence, many current chemotherapeutic agents and radiation therapies depend at least in part on the production of ROS [38]. Previous studies have shown that in tumor cells, radiotherapy-sensitive cells have higher levels of ROS than radiotherapy-resistant cells [7,39]. Likewise, we have demonstrated that NPC cells are less sensitive to IR after reducing intracellular levels of ROS.

Angiotensin-converting enzyme inhibitors have been reported to reduce intracellular ROS levels [40,41,42]. Our findings support the same results. In our study, the ACE inhibitor enalaprilat decreased ROS levels in NPC cells. Consistent with our expectation, enalaprilat reduced IR-induced DNA damage and apoptosis in the NPC cell lines CNE1 and CNE2. Accordingly, our results demonstrate that enalaprilat causes resistance to IR by decreasing the level of ROS in NPC cells.

ACE is a zinc-dependent dipeptidyl carboxypeptidase responsible for converting angiotensin I to angiotensin II [43]. It belongs to the renin-angiotensin system [44]. In fact, the renin-angiotensin system has been found to be highly expressed in tumors from different types of cancers [45]. ACE has been reported to exert antitumor effects as well [46]. 

Recent studies have shown that overexpression of ACE in neutrophils increases intracellular ROS levels [11]. We hypothesized that the increased sensitivity to IR due to ACE overexpression is associated with increased levels of intracellular ROS in tumor cells. To prove this assumption, we analyzed the relationship between intracellular ROS levels and ACE. Our results show that ACE knockdown in NPC cells results in decreased levels of ROS. Conversely, overexpression of ACE in NPC cells increases the level of intracellular ROS in NPC cells and as well as the level of intracellular ROS after IR exposure. It is well known that high ROS levels result in DNA damage [47]. Our results show that ACE knockdown in NPC cells reduces IR-induced DNA damage and apoptosis. Similarly, ACE was overexpressed in the NPC cell lines. We found that ACE overexpression in NPC cells increases IR-induced DNA damage and apoptosis. Thus, we suggest that ACE influences the effect of IR by regulating the level of ROS in NPC cells. 

## 5. Conclusions

In conclusion, our results suggest that ACE inhibition reduces the level of ROS in NPC cells, leading to reduced sensitivity to radiotherapy. Similarly, ACE overexpression enhances the sensitivity of nasopharyngeal carcinoma cells to IR by increasing the level of ROS. 

## Figures and Tables

**Figure 1 biomedicines-11-01581-f001:**
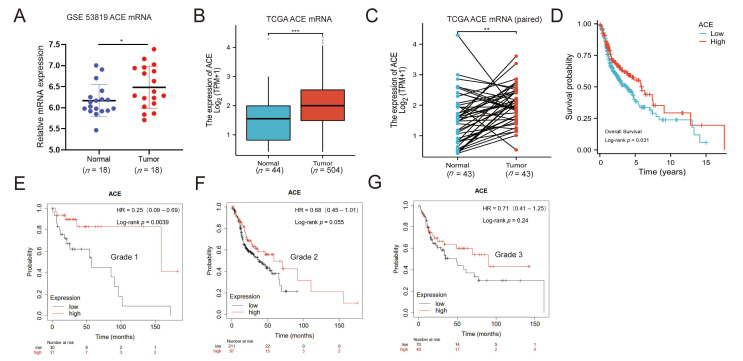
ACE expression is upregulated in NPC patients. (**A**) Expression levels of ACE mRNA in 18 NPC tissue and 18 normal tissue samples from the GEO database (GEO: GSE53819). (**B**) Expression of ACE in HNSCC and normal tissue samples from the TCGA database. (**C**) Comparison of ACE expression levels in 43 pairs of samples in TCGA database HNSCC. (**D**) Kaplan–Meier analysis of overall survival between the high ACE expression group and low ACE expression group in HNSCC. Data were obtained from the TCGA database. (**E**–**G**) Kaplan–Meier curve from HNSCC patients in the KM-Plotter database. Those with higher expression levels of ACE mRNA had better outcomes among Grade 1 patients, but this was not significant for Grade 2 and Grade 3 patients. N.S., *p* ≥ 0.05; *, *p* < 0.05; **, *p* < 0.01; ***, *p* < 0.001.

**Figure 2 biomedicines-11-01581-f002:**
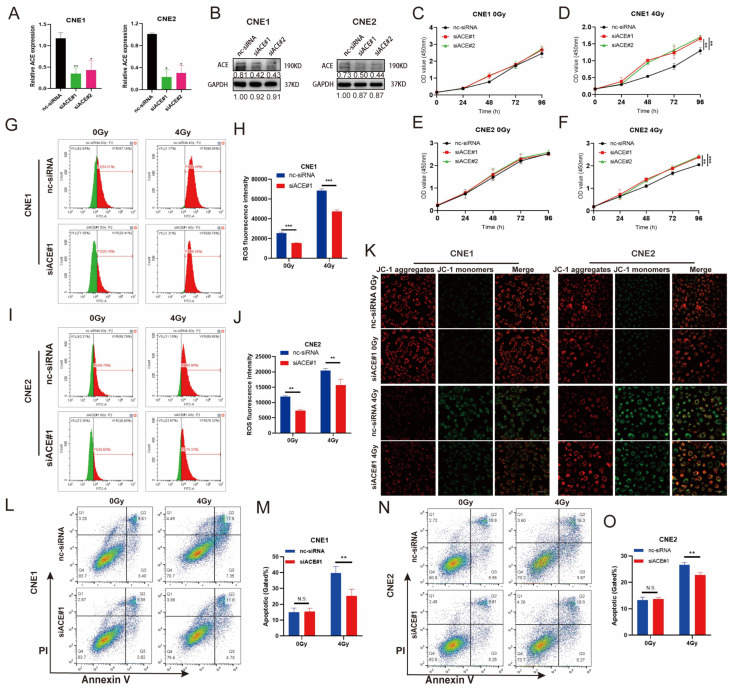
ACE knockdown diminishes the sensitization of NPC cells to IR by reducing the ROS level. (**A**) The mRNA level of ACE was measured by RT-qPCR after ACE knockdown. (**B**) The protein level of ACE was detected by WB after ACE knockdown. (**C**–**F**) The viability of NPC cells was measured by CCK8 assay. ACE knockdown via siRNA treatment and irradiation with a dose of 4 Gy radiation shows sustained growth promotion, although ACE knockdown alone does not increase the proliferation of these cells. (**G**–**J**) The mean fluorescence intensity of DCFH-DA (DCF) was measured using flow cytometry to determine the level of intracellular ROS. Green: fluorescence histogram on the left side of the scale line; red: fluorescence histogram on the right side of the scale line. ROS levels were reduced after ACE knockdown compared to controls in both the IR and non-IR groups. (**K**) Mitochondrial membrane potential changes measured by laser confocal microscopy (JC-1 staining). ACE knockdown reverses radiotherapy-induced decrease in mitochondrial membrane potential in NPC cell lines. Red, JC-1 aggregates; green, JC-1 monomers; merge, combined red and green. Scale bar in the figure is 40 µm. (**L**–**O**) Apoptosis rates determined by flow cytometry. ACE knockdown decreased apoptosis induced by radiotherapy, although ACE knockdown alone did not significantly induce apoptosis. All data are presented as mean ± SEM (*n* = 3). N.S., *p* ≥ 0.05; *, *p* < 0.05; **, *p* < 0.01; ***, *p* < 0.001.

**Figure 3 biomedicines-11-01581-f003:**
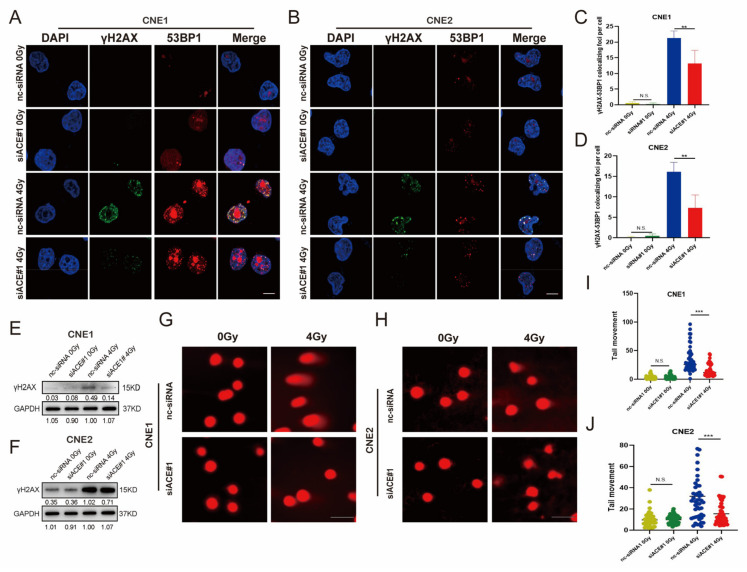
ACE knockdown reduces IR-induced DNA damage. (**A**–**D**) Immunofluorescence staining of NPC cells 2 h after IR exposure. Cells were stained with antibodies against 53BP1 (red) and γH2AX (green). Nuclear DNA was visualized by DAPI (4’,6-diamidino-2-phenylindole, blue). ACE knockdown observed fewer H2AX-53BP1 colocalization foci after 4Gy IR exposure (50 cells counted per group, scale bar 10 μm). (**E**,**F**) γH2AX was detected by western blot. GAPDH (Glycoaldehyde-3-phosphate dehydrogenase) was used as a loading control. (**G**–**J**) Comet assay to detect DNA damage in NPC cells after IR. ACE knockdown did not change the comet tail moment; however, ACE knockdown significantly reduced the comet tail moment after 4Gy IR exposure (counting 50 cells, scale bar 100 µm). All data are presented as mean ± SEM (*n* = 3). N.S., *p* ≥ 0.05; **, *p* < 0.01; ***, *p* < 0.001.

**Figure 4 biomedicines-11-01581-f004:**
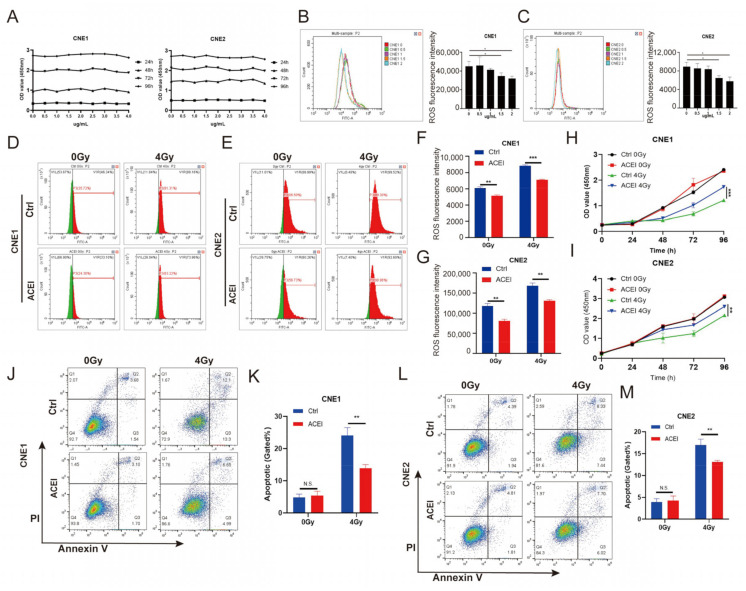
Inhibition of ACE diminishes the sensitization of NPC cells to IR by reducing the ROS level. (**A**) Cell proliferation was measured by CCK8 assay at 24, 48, 72, and 96 h after treatment with enalaprilat. (**B**,**C**) ROS content was measured by flow cytometry after cells were treated with enalaprilat. (**D**–**G**) The mean fluorescence intensity of DCFH-DA was measured using flow cytometry to determine the level of intracellular ROS. Green: fluorescence histogram on the left side of the scale line; red: fluorescence histogram on the right side of the scale line. ROS levels were reduced after ACE inhibition compared to controls in both the IR and non-IR groups. (**H**,**I**) The viability of NPC cells was measured by CCK8 assay. In NPC cell lines, inhibition of ACE did not affect cell proliferation; however, ACE inhibition promoted nasopharyngeal carcinoma cell proliferation after 4Gy IR exposure. (**J**–**M**) Apoptosis rates determined by flow cytometry. Inhibition of ACE decreased apoptosis induced by radiotherapy, although ACE inhibition alone did not significantly induce apoptosis. All data are presented as mean ± SEM (*n* = 3). N.S., *p* ≥ 0.05; *, *p* < 0.05; **, *p* < 0.01; ***, *p* < 0.001.

**Figure 5 biomedicines-11-01581-f005:**
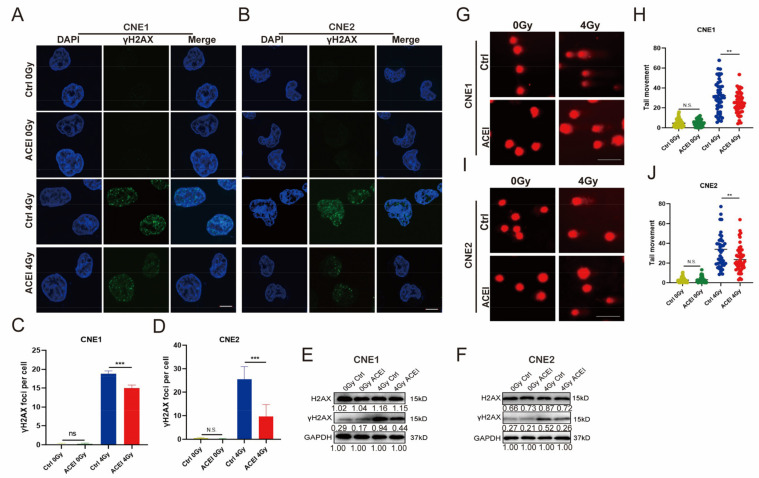
Inhibition of ACE reduces IR-induced DNA damage in NPC cells. (**A**–**D**) Immunofluorescence assays were used to determine γH2AX foci (γH2AX foci, green). Nuclear DNA was visualized by DAPI. The frequencies of total γH2AX foci were counted by Image J software (50 cells counted per group, scale bar 10 μm). (**E**,**F**) Western blot analysis of the expression of γH2AX in NPC cells treated with or without enalaprilat at 2 h after IR. GAPDH was used as a loading control for western blot. (**G**–**J**) Comet assay of NPC cells treated with or without EP after IR (50 cells counted per group, scale bar 100 μm). All data are presented as mean ± SD (*n* = 3). N.S., *p* ≥ 0.05; **, *p* < 0.01; ***, *p* < 0.001.

**Figure 6 biomedicines-11-01581-f006:**
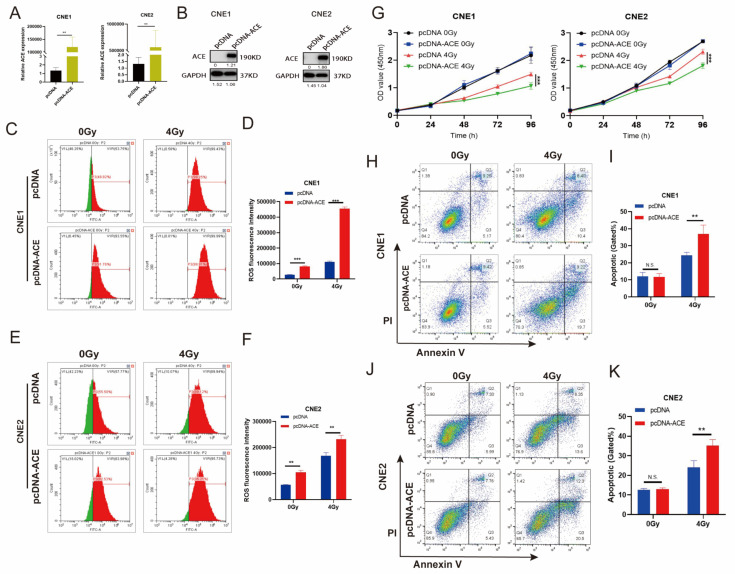
ACE overexpression increases the level of ROS in NPC cells, leading to enhanced sensitivity to radiotherapy. (**A**) The mRNA level of ACE was measured by RT-qPCR after ACE overexpression. (**B**) The protein level of ACE was detected by WB after ACE overexpression. (**C**–**F**) The mean fluorescence intensity of DCFH-DA was measured using flow cytometry to determine the level of intracellular ROS. Green: fluorescence histogram on the left side of the scale line; red: fluorescence histogram on the right side of the scale line. ROS levels were increased after ACE overexpression compared to no overexpression in both the IR and non-IR groups. (**G**) The viability of NPC cells was measured by CCK8 assay. ACE overexpression and irradiation with a dose of 4 Gy showed sustained growth inhibition, although ACE overexpression alone did not reduce the proliferation of these cells. (**H**–**K**) Apoptosis rates determined by flow cytometry. ACE overexpression increased apoptosis induced by radiotherapy, although ACE overexpression alone did not significantly induce apoptosis. All data are presented as mean ± SD (*n* = 3). N.S., *p* ≥ 0.05; **, *p* < 0.01; ***, *p* < 0.001.

**Figure 7 biomedicines-11-01581-f007:**
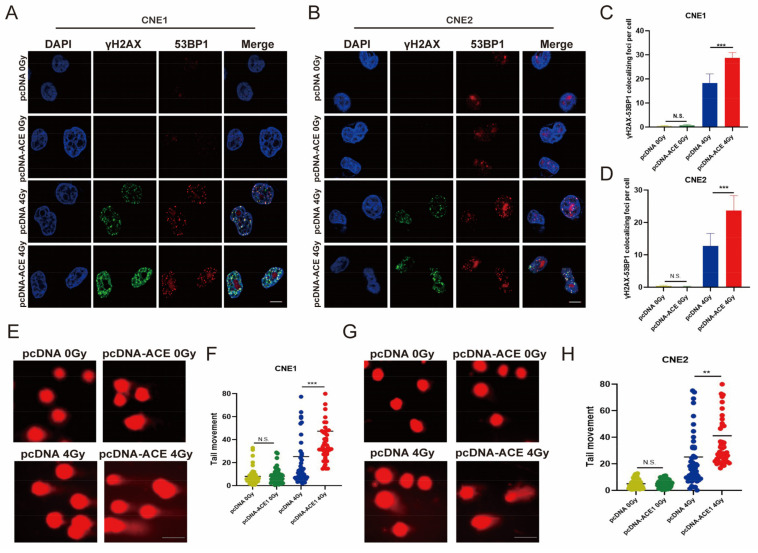
ACE overexpression enhances IR-induced DNA damage. (**A**–**D**) Immunofluorescence staining was performed after IR to measure DNA damage. Cells were stained with antibodies against 53BP1 (red) and γH2AX (green). Nuclear DNA was visualized by DAPI. In the ACE overexpression group, more H2AX-53BP1 colocalization foci were observed after 4Gy IR (50 cells counted per group, scale bar 10μm). (**E**–**H**) Comet assay showing DNA damage detected in NPC cells after IR. In the NPC cell lines CNE1 and CNE2, ACE overexpression did not alter the comet tail moment; however, ACE overexpression significantly increased the comet tail moment after 4Gy IR exposure (counting 50 cells, scale bar 100 μm). All data are presented as mean ± SD (*n* = 3). N.S., *p* ≥ 0.05; ***, *p* < 0.001.

## Data Availability

The data presented in this study are available on request from the corresponding author.

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
