# Peer review of "Learning and Investigation of the Role of Angiotensin-Converting Enzyme in Radiotherapy for Nasopharyngeal Carcinoma"

_biomedicines, 2023, doi:10.3390/biomedicines11061581_

Round 1
Reviewer 1 Report
First of all the title of the MS misguided readers because cognitive learning at first related to neurology/psychiatry .Better to use something like in silico analalysis
I have no doubts that authors accurately quantified ACE mRNA in NPC samples and made correct Kaplan-Meier survival analysis .
However I have doubts about physiological meaning of this mRNA.
Using PCR you can quantify 5 molecules of ACE mRNA per cell and any manipulations with this mRNA has no sense, because ACE mRNA in given cells has physiological meaning if amount of ACE is significant. For example, freshly isolated human endothelial cells expressed significant amount of ACE -about 25 mU/mg of cell protein, but after several passages ACE expression will decrease 5-times and after 10th passages residual ACE will be 5% from initial, but ACE mRNA could be quantified even after 20th passages, but this ACE will not have physiological sense.
By other words, in 2023 it is not enough to measure ACE mRNA, it is necessary to understand how much ACE protein expressed in these cells in comparison with cells where ACE participate in signaling or physiological cascades.
Especially it is important because at the end of this MS authors performed overexpression of ACE in CNE1 and CNE2 lines at it seems to me that the level of protein ACE expression in these cells will be 1000-times higher than in original cells from patients.
In any cases, when physiological experiments performed on different cells or tissues it is necessary to understand with what range of protein of interest (in your case it is ACE) you are dealing with.
The last concern: you are using final concentration of enalaprilat at 10 uM
which is 100-200 fold more than clinical concentration of ACE inhibitors.
It is a usual practice to use for in vitro higher concentrations of the reagents,
but t least you should mentioned (discussed) it.
English of the MS should be improved :
line 482: Similarly, ACE was overexpression in the NPC cell lines
Reviewer 2 Report
Angiotensin-converting enzyme (ACE) inhibitors have been reported to ameliorate morbidities associated with radiation therapy (RT). Here, the role of ACE in modulating nasopharyngeal carcinoma (NPC)response to RT is evaluated using two preclinical models of NPC and in vitro studies. Head and neck cancer publicly available patient tumor, “normal tissue” and survival data from the TCGA were interrogated for ACE mRNA levels in tumor and “normal tissue” and survival data by ACE mRNA level high versus low were evaluated stratified by disease stage. Limitations of the studies are the lack of an in vivo model. All experiments are cell culture experiments. Therefore, the focus of the report is cell intrinsic response to RT. It is unclear why many references are immunological when no immune studies are presented.
Overall, the manuscript is well written and logically presented. The authors demonstrate that ACE has a role in ROS and DNA damage generation and apoptosis using knock-down and overexpression experiments in NPC cell lines. Some of the conclusions are overstated, especially as relates to the interrogation of the TCGA data analysis. Some additional issues are listed below.
1. The TCGA clinical annotation data set provides information regarding adjuvant treatment. Some patients likely received RT as part of treatment. Would be of interest to perform a subset analysis on these patients and assess association with ACE expression level and survival in this subset. This may shed light on the finding of high ACE levels being associated with better prognosis in early stage disease while not being significantly associated in more advanced disease.
2. Please provide axes labels in flow cytometry figures 2L and 2N. It is unclear which gate(s) are being used to define the apoptotic cells, presumably this is the annexin V positive populations, but this should be specifically defined.
3. Define ACEI in Figure 4 legend, such as enalaprilat (ACE, if this is indeed correct.
4. Figure 5 legend: EP needs to be defined as does ACEI. Aren’t both abbreviations for enalaprilat? Abbreviations should be normalized.
5. Line 429-430, the authors state “these finding suggest that ACE induces radiotherapy resistance in NPC cells by modulating the level of ROS.” This conclusion seems incorrect. Data presented support the conclusion that ACE induces radiotherapy sensitization and reduction of ACE level promotes increased ROS and increased DNA damage.
6. Lines 460-461, the authors write the following: “Therefore, we performed a health informatics assessment of ACE causing radiotherapy resistance in nasopharyngeal carcinoma.” The evaluation of the TCGA dataset did not address radiotherapy resistance nor specifically nasopharyngeal cancer. This statement needs to be revised for accuracy.
7. It is unclear what is meant by the authors in the concluding sentence: “Our findings provide a reference for radiotherapy treatment of patients with nasopharyngeal carcinoma.” Recommend this sentence be more specific or be omitted.
Minor
1. Line 210 should read “Grade 2 and Grade 3 patients.”
2. Line 224 should read “…does not increase the proliferation …”
Reviewer 3 Report
The authors found that knockdown or inhibition of ACE in NPC cells reduced IR-induced ROS levels, IR-induced DNA damage and apoptosis. Overexpression of ACE increased ROS levels, DNA damage, and apoptosis in NPC cells.
This paper reported that ACE is involved in radiosensitivity in NPC treatment and may contribute to understanding the behavior of NPCs in radiotherapy. However, there are some points to reconsider, as described below.
The authors noted that ACE is involved in the amount of ROS produced by RT in NPC. However, the mechanism by which ACE promotes ROS production is unknown. It needs to be explained.
The authors need to explain how the findings of this study will contribute to improved radiotherapy with IR for NPC.
As stated by the authors, IR induced DNA damage both directly and indirectly in treated cells. ROS generated by the indirect pathway induce DNA damage. Therefore, in order to understand the direct effects of IR on DNA, experiments using ROS scavengers such as NAC (N-acetyl-L-cysteine) and catalase are required to minimize the indirect effect of IR.
The authors determine ROS levels at 24 h after IR treatment, while gH2AX and 53BP1 expression and comet assays were performed at 2 h after IR treatment. The ROS produced at 24 h could damage DNA. The authors need to explain why they measure ROS and DNA damage at these time points.
Figure 1: It is unclear how Figure 1D differs from Figure 1E.
ACE was detected in CNE1 and CNE2 cells (Fig, 2A, B), but not in Fig. 6B. This discrepancy needs to be explained. Do radioresistant cells express less ACE?
Expression of gH2AX was induced 2 h after 4G of IR (fig. 3). Is ACE expression also affected by 4G of IR?
In flow cytometry, it is necessary to specify which populations are counted as apoptotic cells.
Page 7: Lines 244-248 are duplicate sentences.
None
Reviewer 4 Report
The authors described that angiotensin-converting enzyme (ACE) influences the effect of ionizing radiation on nasopharyngeal carcinoma cells via decreasing the ROS level and enhancing radiation-induced DNA damage and apoptosis. The experimental methods used, and data are very clear and appropriate. The topic addressed is interesting and deserves a constructive discussion. Reviewer is asked to make a few minor corrections and recheck the text.
Comments:
1) In Materials and Methods, on line 85, CCK reagent (Dojindo, Kumamoto, Japan) is correct country.
2) Reviewer would like to see a clear indication of what the X-Y axis title is in figure 2L and 2N. Perhaps annexin V and PI, though.
3) In figure 1E to 1H, the Kaplan-Meier curves are small and unclear, especially “number at risk” as below of X axis. Please describe them more legibly.
4) Line238-243 and line244-248: The exact same sentence is repeated twice, please correct it.
5) In figure 3E-F, authors have measured the brightness of the protein bands, why not normalized the with GAPDH as positive control and calculate the gammaH2AX values for comparison?
6) In figure 4A, the data for the “12-hour incubation” written on line 307, is not shown. Do you have “12-hour” data or not?
7) In figure 4B-4C, the histograms are small and unclear. The different colors of lines are not legible.
8) Regarding the mitochondria membrane potential by JC-1, the results of fluorescent staining are obvious. Do you have quantitative data of MMP? We think it would be more convincing.
9) In figure 5, DNA damage by IR is suppressed by the addition of enalaprilat, although we do not see ROS in particular. Nevertheless, what is the “reducing intracellular reactive oxygen species” in line368 and the title of Fig 5? Isn’t it unnecessary?
10) Authors described that “ACE induces radiotherapy resistance in NPC cells” on line 430. On the other hand, authors said that “ACE can enhance radiosensitivity in nasopharyngeal carcinoma cells” on line484. Are the meanings of the two sentences contradictory? Do these sentences explain discrepancies?
I would like to see a detailed review again of the mixing of abbreviated and non-abbreviated words and presentations that may be a bit confusing to the readers.
1) About CCK8 assay: As written in Materials and Methos, and figure legend, CCK8 assay should be capitalized and unified, not cck8.
2) Throughout, reactive oxygen species or ROS are used intermixedly with and without abbreviation; what is the point of distinguishing between official name (without abbreviation) and ROS (with abbreviation)?
3) Same as comment no.6, nasopharyngeal carcinoma and NPC are used intermixedly with or without abbreviation. What is the intent?
4) At line 329, 2 ug/mL is uncorrected. Microgram/mL is correct.
Round 2
Reviewer 1 Report
I understood that you had no possibility to estimate ACE protein in cells which data on mRNA was in database
Reviewer 3 Report
The authors appropriately respond to most of the points raised by the reviewer.
